# Prognostic Factors of Early Stage Epithelial Ovarian Carcinoma

**DOI:** 10.3390/ijerph16040637

**Published:** 2019-02-21

**Authors:** Shu-Feng Hsieh, Hei-Yu Lau, Hua-Hsi Wu, Heng-Cheng Hsu, Nae-Fang Twu, Wen-Fang Cheng

**Affiliations:** 1Department of Obstetrics and Gynecology, College of Medicine, National Taiwan University, Taipei 100, Taiwan; shuhsieh02421@yahoo.com (S.-F.H.); b101092037@gmail.com (H.-C.H.); 2Department of Obstetrics and Gynecology, Taipei Veterans General Hospital, Taipei 111, Taiwan; belindalau31@gmail.com (H.-Y.L.); hswu@vghtpe.gov.tw (H.-H.W.); 3Department of Obstetrics and Gynecology, National Taiwan University Hospital Hsin-Chu Branch, Hsin-Chu City 300, Taiwan; 4Graduate Institute of Clinical Medicine, College of Medicine, National Taiwan University, Taipei 100, Taiwan; 5Graduate Institute of Oncology, College of Medicine, National Taiwan University, Taipei 100, Taiwan

**Keywords:** ovarian cancer, platinum, taxane, histologic type, disease-free survival, overall survival

## Abstract

We aimed to determine prognostic factors of early stage (I/II) epithelial ovarian carcinoma (EOC) including clinicopathologic and chemotherapeutic regimens. Four hundred and thirty-seven women who underwent primary staging surgery with adjuvant chemotherapy between January 1, 2000 and December 31, 2010 were retrospectively reviewed and analyzed from two medical centers. The prognostic factors were determined from multivariate survival analyses using Cox regression models. The majority of women were diagnosed with stage Ic (244/437, 55.8%). The histopathologic types were clear cell (37.5%), endometrioid (27.2%), serous (14.0%), and mucinous (13.3%). Fifty-seven percent (249/437) of the women received taxane-based (platinum plus paclitaxel) regimens and 43.0% received non-taxane (platinum plus cyclophosphamide) regimens as frontline adjuvant chemotherapy. Clear cell tumors (adjusted Hazard ratio (aHR) 0.37, 95% confidence interval (CI) 0.21–0.73, *p* = 0.001) showed better 5-year disease-free survival (DFS) than serous tumors. Women diagnosed at FIGO (International Federation of Gynecology and Obstetrics) stage II (aHR 5.97, 95% CI = 2.47–14.39, *p* < 0.001), grade 3 tumor without clear cell (aHR 2.28, 95% CI = 1.02–5.07, *p* = 0.004) and who received 3–5 cycles of non-taxane regimens (aHR 3.29, 95% CI = 1.47–7.34, *p* = 0.004) had worse 5-year overall survival (OS). Clear cell histology treated with taxane-based regimens showed significantly higher 5-year DFS (91.2% vs. 82.0%, aHR = 0.45, 95% CI = 0.21–0.93, *p* = 0.043) and 5-year OS (93.5% vs. 79.0%, aHR = 0.30, 95% CI = 0.13–0.70, *p* = 0.005) than those treated with non-taxane-based regimens. We conclude that stage, tumor grade, and chemotherapeutic regimens/cycles are independent prognostic factors for early stage ovarian cancer.

## 1. Introduction

Epithelial ovarian carcinoma (EOC) is the most lethal gynecological cancer. The cell type is a useful prognostic factor of EOC [1], although the clinical relevance of the histological type remains uncertain, and findings regarding the influence of cell type on survival outcomes are inconsistent [2]. Ovarian clear cell carcinomas diagnosed at an advanced stage have an unfavorable prognosis [1,3,4], whereas in the early stage, the prognosis of clear cell histology is similar to serous histology [4,5,6,7]. Kobel et al. even indicated that early stage clear cell carcinoma had a better prognosis than serous carcinoma [8].

Conventional adjuvant chemotherapy was considered less beneficial for patients with FIGO (International Federation of Gynecology and Obstetrics) Ia or Ib, non-clear-cell histological types, well-differentiated cells, and for those who underwent optimal surgery [9,10,11]. The long-term follow-up of ICON 1 trial confirmed the efficiency of adjuvant chemotherapy for early stage ovarian cancer, and the benefit was more significant in the high risk group (i.e., FIGO stage Ib/Ic with grades 2/3, any FIGO stage I with grade 3, clear cell histological type, or FIGO stage II) [12]. 

Platinum-based combinational chemotherapy was recommended for EOC. The ICON1 and ACTION two trials showed that platinum-based chemotherapy showed a better response and survival than observation alone after surgery [9,10]. Subsequently, platinum plus cyclophosphamide became the most common front-line chemotherapeutic regimen in EOC [13]. The superiority of taxane in combination with platinum, compared with platinum with cyclophosphamide, was supported by several phase III trials on advanced stage EOC [14,15,16]. Relative to cisplatin, carboplatin has less neurologic toxicity and side effects, and demonstrated comparable effectiveness [16]. Carboplatin with paclitaxel is currently considered the standard first-line adjuvant therapy for EOC [17,18]. However, due to the high cost of paclitaxel, the carboplatin-paclitaxel regimen is not used throughout the world [19,20]. Furthermore, cost-effectiveness is still an important index for evaluating clinical trials [21]; thus, further optimization and evaluation of platinum with cyclophosphamide is still needed, especially in early stage ovarian cancer.

In the present study, we analyzed the factors which influence the outcome of early stage ovarian carcinoma such as stage, histologic type, and tumor grade; first-line adjuvant chemotherapeutic regimens; and the cycles of chemotherapy.

## 2. Patients and Methods

### 2.1. Patients

The medical records of women diagnosed with EOC at two medical centers in Northern Taiwan, National Taiwan University Hospital (NTUH) and Taipei Veterans General Hospital (TVGH), between 1 January 2000 and 31 December 2010, were retrospectively reviewed. The study protocol was approved by the Institutional Review Boards of the two hospitals. Patients were eligible if they achieved the following criteria: (1) diagnosed at an early stage (stage I or II), (2) treated with primary cytoreductive surgery including total hysterectomy and bilateral salpingo-oophorectomy or fertility-sparing surgery with unilateral salpingo-oophorectomy alone, infracolic omentectomy, pelvic and/or para-aortic lymph node dissection, and pelvic tumor excision, if needed, (3) the histologic data were reviewed by proficient pathologists who specialize in gynecologic oncology, and (4) had sufficient clinicopathological and survival data regarding the disease prognosis. After surgery, the women with early stages received three to six courses of adjuvant chemotherapy, except those with stage IA and grade I disease. Fertility-sparing surgery to preserve the uterus and/or unilateral ovary was performed for some patients who had not completed their family planning. 

### 2.2. Data Collection

The demographic and clinical data were extracted from individual medical records and stored in a database; this included the age at diagnosis, tumor histology and grade, FIGO stage, type of chemotherapy, and follow-up data. Histological grading and disease staging were based on the FIGO classification [22,23]. The regimens of front-line chemotherapy were classified into platinum-paclitaxel based (PT: carboplatin + paclitaxel/cisplatin + paclitaxel/carboplatin + docetaxel) and cyclophosphamide + caboplatin/cyclophosphamide + cisplatin (CP) groups. The choice of chemotherapeutic regimens with platinum and cyclophosphamide or platinum and paclitaxel was made by the patients based on the physician’s suggestions and the patients’ financial condition because patients did not get reimbursed for paclitaxel by the national health insurance in Taiwan. For the decision of chemotherapeutic cycles, 3 to 6 cycles are recommended by the guidelines of TGOG (Taiwan gynecologic oncologic group) of NHRI (National Health Research Institute) of Taiwan based on the physician’s suggestion and patients’ tolerability to the side effects of chemotherapeutic regimens. Recurrence was confirmed by radiology, physical findings, an elevated serum carcinoma antigen 125 (CA-125) or carcinoembryonic antigen (CEA) level (≥2-fold the upper normal limit) in two consecutive tests within 2-week intervals, or by using tissue from a biopsy.

### 2.3. Statistical Analysis

The OS was considered as the time from the date of primary cytoreductive surgery to the date of the last recorded clinical visit or death from any cause; disease- free survival (DFS) was defined as the time interval from the last date of chemotherapy to clinically defined recurrence, death from any cause, or the last recorded clinical visit, whichever occurred first. Univariate and multivariate Cox-proportional hazard regression models were used to examine the effects of each pathological feature and regimens of front-line chemotherapy on recurrence and survival of the patients. The standard log-rank test was applied to evaluate the trend of an independent variable. Analyses were performed using SPSS 22 (SPSS, Inc., Chicago, IL, USA). All statistical tests were two-sided, and *p* values of <0.05 were considered statistically significant.

### 2.4. Details of Ethics Approval

This study was approved by the Research Ethics Committee at the National Taiwan University Hospital (201310006RIND) and is registered in the ClinicalTrials.gov Protocol Registration System Identifier (NCT03019315). Data cannot be shared publicly because all of the patient data were fully anonymized before we accessed them, and the Research Ethics Committee waived the requirement for informed consent. Data are available from the cancer registries of National Taiwan University Hospital and Taipei Veteran General Hospital after the approval of the Research Ethics Committee of the respective hospital to meet the criteria for access to confidential data.

## 3. Results

### 3.1. Patent Characteristics 

A total of 437 women with early stage EOC met the inclusion criteria, 248 were from NTUH and 189 from TVGH. The demographic and clinicopathologic characteristics of these patients are presented in Appendix A. The median age at diagnosis was 50 years (23–84 years). More than half the women (53.5%, 234/437) were ≤50 years of age. The majority of these women were diagnosed at stage Ic (244/437, 55.8%). The histopathologic types of these 437 patients were clear cell (37.5%), endometrioid (27.2%), serous (14.0%), and mucinous (13.3%) and 56.5% (247/437) had a high tumor grade (grade 3). None of the patients had gross residual tumor after surgery. There were 26 (5.9%) of the 437 patients who received fertility-sparing surgery to preserve their uterus. Fifty-seven percent (249/437) of the women received platinum-based anti-neoplastic drugs plus taxane (PT) regimens and 43.0% received platinum-based anti-neoplastic drugs plus cyclophosphamide (CP) regimens as frontline adjuvant chemotherapy. The average follow-up period for all patients was 7.16 years (0.1–15.8). The 5-year recurrent and cancer-related death rates were 22.1% (94/427) and 15.0% (65/432), respectively.

### 3.2. Analyses of Prognostic Factors for 5-year DFS in Early Stage EOC Women

As shown in Table 1, the univariate Cox regression model indicated that the FIGO stage, histologic type, and tumor grade are significant prognostic factors of 5-year DFS. After adjusting for the association between these factors, FIGO stage Ic (adjusted Hazard ratio (aHR) 1.98, 95% confidence interval (CI) = 1.01–3.89, *p* = 0.043; II: aHR 3.26, 95% CI = 1.75–8.65, *p* = 0.002), tumor grade 3 (aHR 3.89, 95% CI = 1.75–8.64, *p* = 0.001), and three to five cycles of the CP regimen (aHR 2.22, 95% CI = 1.18–4.17, *p* = 0.013) were factors for poor prognosis, when compared with stage Ia/Ib, histologic grade 1, and the six-cycle PT regimen, respectively. In addition, patients with a clear cell histology (aHR 0.37, 95% CI 0.21–0.73, *p* = 0.001) showed better 5-year DFS than those with the serous type by multivariate analysis. Patients who received six cycles of the CP regimen (aHR 0.84, 95% CI = 0.49–1.43, *p* = 0.579) demonstrated similar 5-year DFS to patients who received six cycles of the PT regimen.

### 3.3. Analyses of Prognostic Factors for 5-year OS in Early Stage EOC Women

As shown in Table 2, the univariate Cox regression model revealed that FIGO stage was the only significant prognostic factor of 5-year OS. After adjusting for the association between these factors, FIGO stage, tumor grade, and the scheme of adjuvant chemotherapy significantly correlated with the 5-year OS of early stage EOC. Patients diagnosed at FIGO stage II (aHR 5.97, 95% CI = 2.47–14.39, *p* < 0.001), tumor grade 3 (aHR 2.77, 95% CI = 1.12–6.83, *p* = 0.027), with three to five cycles of the CP regimen (aHR 3.29, 95% CI = 1.47–7.34, *p* = 0.004) had worse 5-year OS compared with those diagnosed at FIGO stage Ia/Ib, tumor grade I, or treated with the six-cycle PT regimen. The 5-year OS did not differ between patients treated with the six-cycle CP vs. PT regimen (aHR 1.38, 95% CI = 0.74–2.58, *p* = 0.309).

### 3.4. Patients with the Clear Cell Histologic Type Demonstrated the Best 5-year DFS but Similar 5-year OS Compared with the Other Histologic Types

After adjusting for the co-effects of FIGO stage, tumor grade, chemotherapeutic regimens, and age at diagnosis, the 5-year DFS and OS curves were constructed for the histologic types (Figure 1). Patients with clear cell histology had the best 5-year DFS rate (88.3%) compared to those with endometrioid (74.4%, aHR = 0.45, 95% CI = 0.21–0.98, *p* = 0.038), mucinous (72.3%, aHR = 0.35, 95% CI = 0.16–0.79, *p* = 0.014), or serous (71.5%, aHR = 0.37, 95% CI = 0.21–0.73, *p* = 0.001) types by log-rank test (Figure 1A). There was no statistical difference in the 5-year DFS among patients with endometrioid, mucinous, and serous types of EOC (*p* = 0.177) (Figure 1A). There was also no statistical difference in the 5-year OS among patients with different histologic types (*p* = 0.11, by log rank test) (Figure 1B). Patients with the mucinous type demonstrated a worse 5-year OS rate (81.1%) than patients with the other histologic types, although the aHRs were not statistically significant (Figure 1B). 

### 3.5. PT (Platinum and Paclitaxel) Regimens Had Better DFS and OS in Patients with Clear Cell Histology than CP (Platinum and Cyclophosphamide) Regimens

The 5-year DFS and OS curves of patients receiving PT and CP chemotherapeutic regimens were first evaluated. The PT groups showed similar 5-year DFS (79.8% vs. 78.2%, aHR = 0.92, 95% CI = 0.61–1.38, *p* = 0.678) (Figure 1C) but better 5-year OS (89.3% vs. 82.7%, aHR = 0.59, 95% CI = 0.36–0.57, *p* = 0.038) (Figure 1D). The 5-year DFS and OS curves of patients receiving PT and CP chemotherapeutic regimens, according to histologic type, are shown in Figure 2. In patients with clear cell histology, PT regimens showed significantly higher 5-year DFS than CP regimens (91.2% vs. 82.0%, aHR = 0.45, 95% CI = 0.21–0.93, *p* = 0.043, log-rank test) (Figure 2A). There was no statistically significant difference in the 5-year DFS between PT and CP treatments in patients with the other histologic types (Figure 2A). In addition, PT regimens demonstrated better 5-year OS than CP regimens in patients with a clear cell histology (93.5% vs. 79.0%, aHR = 0.30, 95% CI = 0.13–0.70, *p* = 0.005, log-rank test). In patients with the serous, endometrioid, or mucinous type, there was no statistically significant difference in the 5-year OS between the PT and CP regimens (Figure 2B).

### 3.6. The PT and CP Regimens Showed Similar DFS and OS in Early Stage Ovarian Cancer Patients

The 5-year DFS and OS curves of patients receiving PT and CP chemotherapeutic regimens, according to FIGO stage, are shown in Appendix A. There was no statistically significant difference in the 5-year DFS between PT and CP treatments in patients with stages Ia/Ib (87.1% vs. 92.5%, aHR = 1.76, 95% CI = 0.46–6.64, *p* = 0.407), stage Ic (82.8% vs. 74.7%, aHR = 0.62, 95% CI = 0.36–1.08, *p* = 0.095), and stage II (63.8% vs. 60.7%, aHR = 0.95, 95% CI = 0.46–1.99, *p* = 0.903) by log-rank test (Appendix A). There was no statistically significant difference in the 5-year OS between PT and CP treatments in patients with stages Ia/Ib (90.4% vs. 95.0%, aHR = 2.02, 95% CI = 0.41–10.0, *p* = 0.391), stage Ic (91.5% vs. 83.4%, aHR = 0.49, 95% CI = 0.23–1.03, *p* = 0.061), or stage II (75.3% vs. 61.0%, aHR = 0.56, 95% CI = 0.27–1.19, *p* = 0.132, by log-rank test) either (Appendix A).

### 3.7. PT Regimens Had Benefit on 5-year OS in Patients with Poorly-Differentiated Tumor Cells (Grade 3) 

The 5-year DFS and OS curves of patients receiving PT and CP regimens, according to tumor grade, are shown in Figure 3A,B. After adjusting for FIGO stage, cell type, and age at diagnosis, there was no statistically significant difference in the 5-year DFS between PT and CP treatments in patients with grade 1 and 2 tumor cells (82.0% vs. 84.5%, aHR = 0.70, 95% CI = 0.18–2.63, *p* = 0.675) or grade 3 (78.4% vs. 72.3%, aHR = 0.73, 95% CI = 0.44–1.21, *p* = 0.224) by log-rank test (Figure 3A). However, in patients with grade 3 tumors, PT regimens showed better 5-year OS than PC regimens (87.2% vs. 78.9%, aHR was 0.52, 95% CI = 0.28–0.95, *p* = 0.034, by log-rank test) (Figure 3B). There was no 5-year OS difference in patients with grade I or grade 2 tumors, when treated with PT or CP regimens.

### 3.8. The Patient’s Age at Diagnosis Did not Affect the 5-year DFS and OS of Patients Treated with PT and CP Regimens 

The 5-year DFS and OS curves of patients receiving PT and CP chemotherapeutic regimens, according to the age at diagnosis, are shown in Figure 3C,D. In patients diagnosed at ≤50 years old, PT regimens had similar 5-year DFS for the PT and CP regimens (82.3% vs. 74.9%, aHR 0.68, 95% CI = 0.38–1.22, *p* = 0.202, Figure 3C). A similar phenomenon was also observed in patients diagnosed at >50 years old (78.2% vs. 79.5%, aHR 0.99, 95% CI = 0.55–1.78, *p* = 0.983, Figure 3C). The 5-year OS curves did not significantly differ between ≤50 years old (89.8% vs. 82.9%, aHR 0.58, 95% CI = 0.29–1.18, *p* = 0.135) or >50 years old patients (89.2% vs. 82.6%, aHR 0.60, 95% CI = 0.30–1.18, *p* = 0.141) (Figure 3D).

## 4. Discussion

The cell type is one of the most useful prognostic factors for advanced stage EOC [24]. A clear cell histology had an unfavorable prognosis in advanced stage EOC [25,26], which was ascribed to the low response rates to adjuvant chemotherapy [26]. However, findings from patients with early stage and advanced EOC were conflicting. In early stage EOC, the prognosis of the clear cell type is similar to the serous type [4,5,6,7]. Kobel et al. even reported that the clear cell histology led to a better prognosis than the serous histology in early stage EOC [8]. Our analysis also indicated that early stage clear cell EOC had a significantly better DFS compared with serous EOC, whereas the mucinous type demonstrated worse OS than non-mucinous histological types. Thus, the histologic type is an important prognostic factor in early stage EOC. 

The number of cycles of adjuvant chemotherapy in early stage EOC patients is another debatable issue. The GOG 157 indicated that the relative risk of recurrence was not different for patients who received six vs. three cycles of adjuvant chemotherapy [4]. Patients receiving the platinum plus paclitaxel regimen had a significantly longer PFS and OS than those receiving the platinum plus cyclophosphamide regimen as adjuvant chemotherapy for advanced stage EOC [14,15,22]. However, Gadducci et al. reported that early stage EOC patients had similar outcomes when treated with non-taxane and taxane-based adjuvant chemotherapy [23]. It provides a good example to evaluate if different chemotherapeutic regimens would influence the outcome of early stage ovarian cancer patients in the real world or in countries where paclitaxel is not available to be used in the treatment of early stage ovarian cancer patients, as done in this analysis. Six cycles of adjuvant chemotherapy were planned for all of the patients. However, some of the patients could only receive 3 to 5 cycles of chemotherapy due to their performance status and patients’ compliance. As shown in Table 1 and Table 2, patients who received 3 to 5 cycles of platinum and cyclophosphamide had significantly poorer DFS (HR 2.22, 95% CI 1.18–4.17, *p* = 0.013) and OS (HR 3.29, 95% CI 1.47–7.34, *p* = 0.004) as compared with patients receiving 6 cycles of platinum and paclitaxel. We hypothesized that incomplete treatment or fewer treatment cycles may result in microscopic residual tumor cells. Our analysis supported that taxane-based adjuvant chemotherapy could improve the outcome of early stage EOC patients. However, platinum and cyclophosphamide for 6 cycles could be an alternative regimen in some countries where paclitaxel is not available.

The heterogeneity of ovarian cancer accounts for the different sensitivities to anti-neoplastic drugs. Our results also indicated that low grade (grade 1/2) tumors had better DFS and OS than high grade (grade 3) tumors. High grade tumors treated with PT regimens had better DFS and OS than those treated with CP regimens, whereas lower grade tumors had similar PFS and OS, regardless of whether they were treated with taxane or non-taxane chemotherapy. We hypothesized that paclitaxel could be more effective than cyclophosphamide in treating chemo-resistant tumors. 

Different histologic types had different outcomes when treated with different regimens of anti-neoplastic drugs, especially clear cell histology. Chan et al. reported that the clear cell histologic type had a poor prognosis due to its resistance to platinum-based chemotherapy [4]. Only patients with a clear cell histology treated with taxane regimens had significantly longer 5-year DFS and OS than those treated with non-taxane regimens in this study. The outcome from the chemotherapeutic regimens did not differ for the other histologic types. 

The limitations of this study include retrospective and institutional biases. This study is retrospective, not randomized and prospective. The choice of chemotherapeutic regimen was decided by patients, their family, and physicians; although, paclitaxel is paid by the patients, and patients are reimbursed for cyclophosphamide by the national health insurance. Thus, income may be a factor when making this decision, and the benefit of PT regimens might be attributed to the second association caused by higher socioeconomic status of the PT treatment group, whereas PT regimens only showed superior outcome of clear cell carcinoma, not the other histologic type (Appendix A). Socioeconomic status is still an important factor in the outcome of early stage ovarian carcinoma patients.

The patients recruited in this study were from two medical centers in the same city. It would be worthwhile to evaluate if the same results were observed in medical centers in other cities or parts of the island, or in other countries. This highlights the need for multi-institutional, prospective studies in the future. In conclusion, stage, tumor grade, and chemotherapeutic regimens/cycles are important prognostic factors for early stage ovarian cancer patients. 

## Figures and Tables

**Figure 1 ijerph-16-00637-f001:**
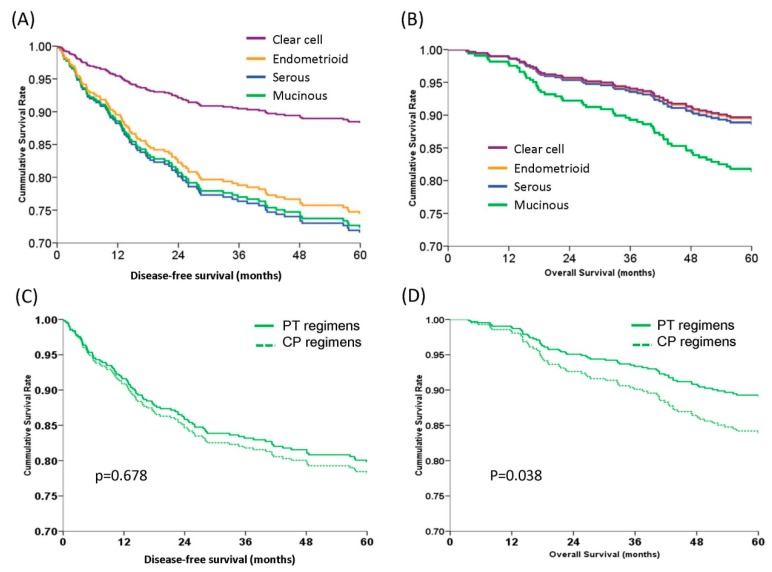
Survival curves of 437 early stage epithelial ovarian carcinoma women with different histologic types or chemotherapeutic regimens by multivariate Cox regression models. (**A**) Estimated 5-year disease-free survival (DFS) curves. The clear cell type (88.3%) demonstrated the best 5-year DFS vs. the endometrioid (74.4%, aHR = 0.45, 95% CI = 0.21–0.98, *p* = 0.038), mucinous (72.3%, aHR 0.35, 95% CI = 0.16–0.79, *p* = 0.014), and serous (71.5%, aHR = 0.37, 95% CI = 0.21–0.73, *p* = 0.001) types by log-rank test. The 5-year DFS rates of patients with endometrioid, mucinous, and serous type did not differ (*p* = 0.177). (**B**) Estimated 5-year overall survival (OS) curves. The mucinous histology showed a trend towards a worse 5-year OS (81.1%) compared with the other histologic types, although it did not reach statistical significance (clear cell 89.2%, *p* = 0.143; endometrioid type 89.1%, *p* = 0.081; serous type 88.4%, *p* = 0.248). (**C**) Estimated 5-year disease-free survival (DFS) curves. There were no differences in the 5-year DFS rates between platinum and paclitaxel (PT) or platinum and cyclophosphamide (CP) regimens (79.8% vs. 78.2%, aHR = 0.92, 95% CI = 0.61–1.38, *p* = 0.678). (**D**) Estimated 5-year overall survival (OS) curves. Patients receiving the PT regimens had better 5-year OS than those receiving the CP regimens (89.3% vs. 82.7%, aHR = 0.59, 95% CI = 0.36–0.57, *p* = 0.038) (all by log-rank test).

**Figure 2 ijerph-16-00637-f002:**
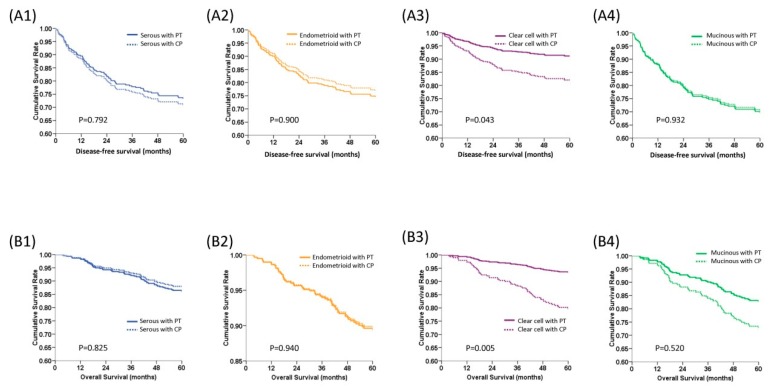
Survival curves of 437 early stage epithelial ovarian carcinoma women stratified by chemotherapeutic regimens and histologic type with multivariate Cox regression models after adjusting for FIGO stage, tumor grade, and age at diagnosis. (**A**) Estimated 5-year disease-free survival (DFS) curves. **A1**: Serous type, **A2**: Endometrioid type, **A3**: Clear cell type, **A4**: Mucinous type. Patients with clear cell histology treated with platinum and paclitaxel (PT) regimens had significantly longer 5-year DFS than those treated with platinum and cyclophosphamide (CP) regimens (91.2% vs. 82.0%, aHR = 0.45, 95% CI = 0.21–0.93, *p* = 0.043, log-rank test). The 5-year DFS of the other histologic types did not differ with PT or CP regimens. (**B**) Estimated 5-year overall survival (OS) curves. **B1**: Serous type, **B2**: Endometrioid type, **B3**: Clear cell type, **B4**: Mucinous. The 5-year OS rate of patients with clear cell histology (93.5% vs. 79.0%, aHR = 0.30, 95% CI = 0.13–0.70, *p* = 0.005) treated with PT regimens was also higher than those treated with CP regimens. There was no difference in 5-year OS, regardless of treatment with PT or CP regimens, in patients with the serous (86.2% vs. 87.8%, *p* = 0.825), endometrioid (89.4% vs. 89.7%, *p* = 0.739), and mucinous (aHR = 0.63, 95% CI = 0.16–2.38, *p* = 0.520) types (all by log-rank test).

**Figure 3 ijerph-16-00637-f003:**
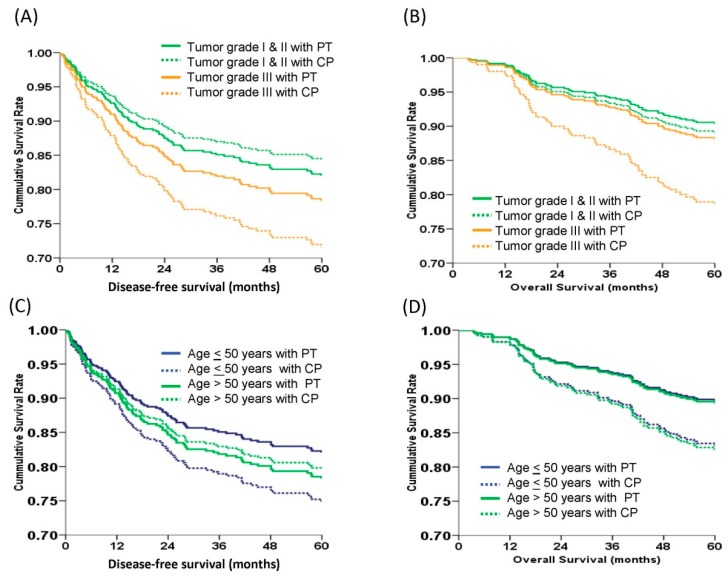
The survival curves of 437 early stage epithelial ovarian carcinoma women stratified by chemotherapeutic regimens and tumor grade or chemotherapeutic regimens and age at diagnosis by multivariate Cox regression models. (**A**) Estimated 5-year disease-free survival (DFS) curves. There was no difference in the 5-year DFS of different tumor grades between PT or CP regimens (grade 1/2 82.0% vs. 84.5%, aHR = 0.70, 95% CI = 0.18–2.63, *p* = 0.675; grade 3 78.4% vs. 72.3%, aHR = 0.73, 95% CI = 0.44–1.21, *p* = 0.224). (**B**) Estimated 5-year overall survival (OS) curves. The 5-year OS rate of patients with grade 3 tumors treated with PT regimens was higher than those treated with CP regimens (87.2% vs. 78.9%, aHR = 0.52, 95% CI = 0.28–0.95, p=0.034). However, the 5-year OS was not different between PT and CP regimens in grade 1/2 tumors (90.2% vs. 88.7%, *p* = 0.76). (**C**) Estimated 5-year disease-free survival (DFS) curves. There was no difference in the 5-year DFS between PT or CP regimens in the two age groups (≤50 years 82.3% vs. 74.9%, aHR 0.68, 95% CI = 0.38–1.22, *p* = 0.202; ≥50 years 78.2% vs. 79.5%, aHR 0.99, 95% CI = 0.55–1.78, *p* = 0.983). (**D**) Estimated 5-year overall survival (OS) curves. The 5-year OS rates of PT and CP were similar in both age groups (≤50 years, 89.8% vs. 82.9%, aHR 0.58, 95% CI = 0.29–1.18, *p* = 0.135; >50 years, 89.2% vs. 82.6%, aHR 0.60, 95% CI = 0.30–1.18, *p* = 0.141) (all by log-rank test).

**Table 1 ijerph-16-00637-t001:** Prognostic factors for 5-year disease-free survival (DFS) of 437 early stage ovarian cancer women by univariate and multivariate Cox regression analyses.

Prognostic Factors	Univariate Regression Analyses			Multivariate Regression Analyses		
	HR	95% CI	*p*	aHR	95% CI	*p*
Age (y/o)						
≤50	1.00	(Reference)		1.00	(Reference)	
>50	1.20	0.80–1.79	0.383	0.98	0.64–1.51	0.935
FIGO stage						
Ia and Ib	1.00	(Reference)		1.00	(Reference)	
Ic	1.92	1.00–3.69	0.049	1.98	1.01–3.89	0.043
II	3.84	1.94–7.63	<0.001	3.26	1.75–8.65	0.002
Histology						
Serous	1.00	(Reference)		1.00	(Reference)	
Mucinous	0.50	0.25–1.00	0.049	0.98	0.45–2.12	0.964
Endometrioid	0.44	0.25–0.79	0.006	0.89	0.45–1.75	0.734
Clear cell	0.37	0.21–0.64	<0.001	0.37	0.21–0.73	0.001
Other types *	0.67	0.31–1.45	0.311	0.75	0.33–1.69	0.484
Tumor grade						
1	1.00	(Reference)		1.00	(Reference)	
2	1.74	0.84–3.60	0.138	2.12	0.96–4.71	0.064
3	2.11	1.11–4.00	0.023	3.89	1.75–8.64	0.001
Regimens and cycles						
PT (6)	1.00	(Reference)		1.00	(Reference)	
PT (3–5)	0.51	0.23–1.12	0.093	0.77	0.33–1.79	0.538
CP (6)	0.77	0.47–1.26	0.297	0.84	0.49–1.43	0.579
CP (3–5)	1.52	0.86–2.68	0.151	2.22	1.18–4.17	0.013

Abbreviations: aHR: adjusted Hazard ratio; y/o: year old; *Other types includes serous mixed mucinous, serous mixed clear cell, mucinous mixed endometrioid, mucinous mixed clear cell, endometrioid mixed clear cell, poorly-differentiated, transitional, hepatoid, and small and squamous cell carcinoma. PT: Paclitaxel + Platinum, CP: Cyclophosphamide + Platinum, CI: confidence interval. Variables in the multivariate Cox regression models were age, FIGO stage, histological type, tumor grade, regimens and cycles, and hospital; adjusted by year of diagnosis and fertility-sparing surgery.

**Table 2 ijerph-16-00637-t002:** Prognostic factors for overall survival of 437 early stage ovarian cancer women by univariate and multivariate Cox regression analyses.

Prognostic Factors	Univariate Regression Analyses			Multivariate Regression Analyses		
	HR	95% CI	*p*	aHR	95% CI	*p*
Age						
≤50	1.00	(Reference)		1.00	(Reference)	
> 50	1.15	0.59–2.26	0.511	0.91	0.54–1.53	0.731
FIGO stage						
Ia and Ib	1.00	(Reference)		1.00	(Reference)	
Ic	1.37	0.62–3.02	0.366	1.53	0.68–3.42	0.302
II	3.67	1.65–8.17	<0.001	5.97	2.47–14.39	<0.001
Histology						
Serous	1.00	(Reference)		1.00	(Reference)	
Mucinous	0.79	0.35–1.78	0.570	2.03	0.79–5.21	0.140
Endometrioid	0.48	0.22–1.02	0.055	1.18	0.48–2.87	0.718
Clear cell	0.56	0.29–1.10	0.094	0.96	0.45–2.04	0.909
Other types *	0.79	0.30–2.07	0.635	1.33	0.49–3.56	0.571
Tumor grade						
1	1.00	(Reference)		1.00	(Reference)	
2	1.21	0.38–3.81	0.934	1.61	0.51–2.95	0.317
3	1.54	0.59–4.09	0.206	2.77	1.12–6.83	0.027
Regimens and cycles						
PT (6)	1.00	(Reference)		1.00	(Reference)	
PT (3–5)	0.81	0.33–1.97	0.645	1.78	0.70–4.61	0.233
CP (6)	1.24	0.70–2.18	0.465	1.38	0.74–2.58	0.309
CP (3–5)	1.86	0.92–3.77	0.084	3.29	1.47–7.34	0.004

Abbreviation: aHR—adjusted Hazard ratio, * Other types include serous mixed mucinous, serous mixed clear cell, mucinous mixed endometrioid, mucinous mixed clear cell, endometrioid mixed clear cell, poorly-differentiated, transitional, hepatoid, small and squamous cell carcinoma. CP: Cyclophosphamide + Platinum, PT: Paclitaxel + Platinum, CI: confidence interval. Variables in the multivariate Cox regression models were age, FIGO stage, histological type, tumor grade, regimens and cycles, and hospital; adjusted by year of diagnosis and fertility-sparing surgery.

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
