# Peer review of "Prognostic Factors of Early Stage Epithelial Ovarian Carcinoma"

_ijerph, 2019, doi:10.3390/ijerph16040637_

Round 1

Reviewer 1 Report

The manuscript by Shu-Feng Hsieh et al. is a retrospective analysis of 437 cases of stage I/II ovarian carcinoma collected in two different medical institutions of Taiwan. The authors investigate the factors influencing outcome in this patient population (stage, histologic type, and tumor grade; first-line adjuvant chemotherapeutic regimens; and the cycles of chemotherapy).

Even if it suffer from the bias that it is retrospective and include pateints from two medical centers from Taiwan, it is potentially interesting. I have  concerns about the istological diagnosis considering that histotype is a prognostic factor in both early and late stage of ovarinan carcinomas. Was the histological diagnosis only retriewed from the data base or were the cases hisytologiclaly reviwed by another pathologist? Particular concer is on the mucinous subtype that could be misdisgnosed with colon carcer metastasis.

All the figures need to be restyled to have the same charater and font (i..e figure 1 and 4)

and Figure 3 can be moved as a supplementary Figure

Author Response

The manuscript by Shu-Feng Hsieh et al. is a retrospective analysis of 437 cases of stage I/II ovarian carcinoma collected in two different medical institutions of Taiwan. The authors investigate the factors influencing outcome in this patient population (stage, histologic type, and tumor grade; first-line adjuvant chemotherapeutic regimens; and the cycles of chemotherapy).

Even if it suffer from the bias that it is retrospective and include pateints from two medical centers from Taiwan, it is potentially interesting. I have  concerns about the histological diagnosis considering that histotype is a prognostic factor in both early and late stage of ovarinan carcinomas. Was the histological diagnosis only retriewed from the data base or were the cases hisytologiclaly reviwed by another pathologist? Particular concer is on the mucinous subtype that could be misdisgnosed with colon carcer metastasis.

All the figures need to be restyled to have the same charater and font (i..e figure 1 and 4) and Figure 3 can be moved as a supplementary Figure

Ans: Thank you for your comments and suggestion.

1.      The histological diagnosis was only reviewed from the data base which was not reviewed by pathologist. We agreed that it is the deficit of retrospective study. However, the diagnosis of all of the patients and their histologic types were reported by gynecologic pathologists of the two hospital. We believe that the misdiagnosis of mucinous type could be decreased as much as possible.  

2.      We have restyled all of the figures by your suggestion. (Please see the revised figures 1to 3 and supplementary figure 1).

3.      Figure 3 was changed to be the supplementary figure 1 by your suggestion. (Please see supplementary figure 1).

Reviewer 2 Report

Interesting study on 437 early-stage OC patients.

The most important concern is about the operative quality. This has to be adressed before finally evaluating this manuscript:

- Have all patients been operated according to the Guidelines?

- Was all tumor removed in ALL patients? Posoperative tumor rest is the most important prognostic factor. This has to be adressed.

- Have patients with fertility sparing treatment been excluded? Otherwise, this should also be considered in analysis.

- Operative strategies must also be considered in uni- or multivariate analysis, especially in early-stage OC!

A clear and straight analysis is urgently necessary to find clear conclusions. Otherwise, it only presents another study with conflicting data.

Additionally: The discussion part should definitely be more precise. Please discuss your results and compare it to the literature instead of talking about possible immunotherapeutic strategies which are not really covered by your paper. Especially, if you focus on low-income countries, immunotherapies will probably not be the solution. 

Author Response

Interesting study on 437 early-stage OC patients.

The most important concern is about the operative quality. This has to be adressed before finally evaluating this manuscript:

- Have all patients been operated according to the Guidelines?

Ans: Thank you for your comments. All of the patients received primary cytoreductive surgery iprimary cytoreductive surgery including total hysterectomy and bilateral salpingo-oophorectomy or fertility-sparing surgery with unilateral salpingo-oophorectomy alone, infracolic omentectomy, pelvic and/or para-aortic lymph node dissection, and pelvic tumor excision if needed, followed by adjuvant platinum-based chemotherapy. (Please see Page 6, Lines 8-12 and 15-18)

- Was all tumor removed in ALL patients? Posoperative tumor rest is the most important prognostic factor. This has to be adressed.

Ans: Thank you for your comments. None of the patients had gross residual tumor after surgery. (Please see Page 9, Lines 10-11)

- Have patients with fertility sparing treatment been excluded? Otherwise, this should also be considered in analysis.

Ans: Thank you for your comments. There were 26 (5.9%) of the 437 patients received fertility-sparing surgery to preserve their uterus. The reasons that we recruited the fertility-sparing surgery patients were that all of these 26 patients also received complete staging surgery and the following statistical analyses were adjusted by fertility-sparing surgery. (Please see Page 9, Lines 11-12)

Operative strategies must also be considered in uni- or multivariate analysis, especially in early-stage OC!

Ans: Thank you for your comments. Actually, the multivariate regression analyses were adjusted by year of diagnosis and fertility-sparing surgery. (Please see Tables 1 and 2)

A clear and straight analysis is urgently necessary to find clear conclusions. Otherwise, it only presents another study with conflicting data.

Ans: Thank you for your comments. We have analyzed our data by a clear and straight analysis to make clear conclusions.

Additionally: The discussion part should definitely be more precise. Please discuss your results and compare it to the literature instead of talking about possible immunotherapeutic strategies which are not really covered by your paper. Especially, if you focus on low-income countries, immunotherapies will probably not be the solution. 

Ans: Thank you for your comments. We have revised the discussion by your suggestion.